# Rosiglitazone Does Not Show Major Hidden Cardiotoxicity in Models of Ischemia/Reperfusion but Abolishes Ischemic Preconditioning-Induced Antiarrhythmic Effects in Rats In Vivo

**DOI:** 10.3390/ph15091055

**Published:** 2022-08-26

**Authors:** Bennet Y. Weber, Gábor B. Brenner, Bernadett Kiss, Tamás G. Gergely, Nabil V. Sayour, Huimin Tian, András Makkos, Anikó Görbe, Péter Ferdinandy, Zoltán Giricz

**Affiliations:** 1MTA-SE System Pharmacology Group, Department of Pharmacology and Pharmacotherapy, Semmelweis University, H-1089 Budapest, Hungary; 2Pharmahungary Group, H-6722 Szeged, Hungary

**Keywords:** rosiglitazone, hidden cardiotoxicity, ischemic preconditioning, I/R injury model, diabetes, thiazolidinedione, simulated ischemia/reperfusion, Avandia

## Abstract

Clinical observations are highly inconsistent with the use of the antidiabetic rosiglitazone regarding its associated increased risk of myocardial infarction. This may be due to its hidden cardiotoxic properties that have only become evident during post-marketing studies. Therefore, we aimed to investigate the hidden cardiotoxicity of rosiglitazone in ischemia/reperfusion (I/R) injury models. Rats were treated orally with either 0.8 mg/kg/day rosiglitazone or vehicle for 28 days and subjected to I/R with or without cardioprotective ischemic preconditioning (IPC). Rosiglitazone did not affect mortality, arrhythmia score, or infarct size during I/R. However, rosiglitazone abolished the antiarrhythmic effects of IPC. To investigate the direct effect of rosiglitazone on cardiomyocytes, we utilized adult rat cardiomyocytes (ARCMs), AC16, and differentiated AC16 (diffAC16) human cardiac cell lines. These were subjected to simulated I/R in the presence of rosiglitazone. Rosiglitazone improved cell survival of ARCMs at 0.3 μM. At 0.1 and 0.3 μM, rosiglitazone improved cell survival of AC16s but not that of diffAC16s. This is the first demonstration that chronic administration of rosiglitazone does not result in major hidden cardiotoxic effects in myocardial I/R injury models. However, the inhibition of the antiarrhythmic effects of IPC may have some clinical relevance that needs to be further explored.

## 1. Introduction

Clinical observations of adverse drug reactions led to the post-market withdrawal of 462 medicinal products from 1953 to 2013 [1]. Among the most common causes of withdrawal was cardiotoxicity [2]. The observation that cardiotoxicities remain undetected during the preclinical and early clinical safety screening can be attributed to the fact that certain cardiac side effects only manifest in the diseased heart, e.g., in ischemia/reperfusion (I/R) injury. Therefore, we defined this phenomenon as “hidden cardiotoxicity” [3].

Hidden cardiotoxicities often manifest as ischemia-related cardiac deaths, I/R-induced arrhythmias, or cardiac dysfunction [3]. Hidden cardiotoxicity of a drug may further deteriorate cell signaling activated by I/R injury or cardiovascular comorbidities. It may also inhibit the cell survival signaling pathways induced by cardioprotective interventions, such as ischemic conditioning [3,4,5,6]. For instance, hidden cardiotoxicity of cycloxygenase-2 inhibitor rofecoxib has been proven to increase mortality due to proarrhythmic effects in a rat model of acute myocardial infarction (MI) [7]. As an example of the interference with protective signaling of ischemic conditioning, Kocsis et al. showed that acute lovastatin treatment abolishes the infarct-size-limiting effect of preconditioning in rats [8].

Preclinical drug safety testing focuses on direct toxicity and adverse drug effects in healthy animal models, thereby overlooking the detection of possible hidden cardiotoxic effects manifesting in I/R and other comorbid conditions [3,9,10]. The current preclinical electrophysiological cardiac safety test guidelines do not advocate testing of proarrhythmic properties in arrhythmia-susceptible tissues or animals [11,12]. Thus, drug toxicity can be concealed when healthy hearts are investigated but may appear in the presence of cardiac diseases, e.g., in myocardial I/R [4,6].

Rosiglitazone, a thiazolidinedione-type antidiabetic and a potent selective agonist of the transcription factor PPARγ (peroxisome proliferator-activated receptor γ) was subject to post-marketing safety concerns of cardiotoxicity [13]. This led to withdrawal from the European market and temporarily placed it under marketing restrictions by the FDA [14,15]. In the open-label RECORD trial (Rosiglitazone Evaluated for Cardiac Outcomes and Regulation of Glycemia in Diabetes study), 4447 patients with type 2 diabetes received rosiglitazone or the standard-of-care treatment, metformin with a sulphonylurea. Results showed an increased risk of heart failure but not cardiovascular death or MI [16]. A cross-sectional study of the Veterans Affairs Diabetes Trial (VADT) further supported these findings by comparing rosiglitazone to the standard-of-care treatment combination [17]. A meta-analysis involving nine randomized, controlled trials compared rosiglitazone add-on therapy to insulin monotherapy and did not find an increased risk for any deleterious cardiovascular outcomes [18]. In contrast, data from two meta-analyses with 14,291 and 27,847 participants, respectively, identified an increased risk for MI compared to placebo or other non-thiazolidinedione oral hypoglycemic drugs [19,20]. One meta-analysis even associated an increased risk of mortality with rosiglitazone due to cardiovascular causes [19]. A recent umbrella review evaluating 232 meta-analyses for ten classes of diabetes drugs confirmed the increased risk for MI compared to its active control. However, no increased risk was found compared to placebo treatment [21].

The fact that the cardiotoxic effects of rosiglitazone remained hidden until post-marketing studies underlines the need to develop safety testing protocols to detect hidden toxicity in early preclinical models with comorbidities. This would reduce the sunk cost of drug development and increase patient safety [3].

Here we aimed to test the possible hidden cardiotoxicity of rosiglitazone in I/R injury models with and without cardioprotection by ischemic preconditioning (IPC).

## 2. Results

### 2.1. In Vivo Model of I/R Injury with IPC

To uncover the hidden cardiotoxic properties of rosiglitazone in vivo, we performed I/R injury experiments in rats (Figure 1).

#### 2.1.1. Chronic Rosiglitazone Treatment Did Not Influence Infarct Size or Interfere with the Infarct-Size-Limiting Effects of Ischemic Preconditioning

First, we aimed to investigate the hidden cardiotoxicity of rosiglitazone by measuring the infarct size after coronary occlusion and reperfusion. Rats were treated for four weeks, with the last dose being administered 24 h before the surgery. As illustrated in Figure 1, the animals were subjected to 30 min of ischemia and 120 min of reperfusion with or without IPC. As the hard endpoint for ischemic injury, infarct size was measured as a proportion of the total tissue exposed to ischemia (i.e., risk area) to explore the effect of rosiglitazone on I/R injury and cardioprotection by IPC. The myocardial area at risk (AAR) is defined as the ischemic proportion of the myocardium that reflects the potential size of the MI after coronary occlusion. Figure 2 shows that infarct size was unaffected by rosiglitazone treatment compared to the I/R vehicle group. Cardioprotection by IPC successfully reduced the infarct size in rosiglitazone- and vehicle-treated animals. In conclusion, rosiglitazone did not affect I/R injury or interfere with the infarct-size-limiting effect of IPC. Neither rosiglitazone nor IPC significantly influenced the AAR (expressed as % of the left ventricle) (I/R vehicle: 43.6 ± 4.7%; I/R rosiglitazone: 48.3 ± 2.3%; IPC vehicle: 37.7 ± 2.9%; IPC rosiglitazone 41.34 ± 2%).

#### 2.1.2. Chronic Rosiglitazone Treatment Abolished the Antiarrhythmic Effect of IPC

According to the Lambeth conventions, cardiac arrhythmias were assigned a score reflecting their severity and accumulation during ischemia and early reperfusion. Figure 3 presents the arrhythmia scores of animals corresponding to the treatment groups illustrated in Figure 1. The score did not significantly increase with rosiglitazone treatment and I/R compared to the vehicle-treated group, nor was there a significant difference between the two IPC groups. Comparing the two vehicle-treated groups showed that IPC successfully protected the animals from arrhythmia development. However, as shown in Figure 3, the results indicate no statistical difference between the two rosiglitazone-treated groups. This finding suggests that rosiglitazone abolishes the antiarrhythmic effect of IPC.

#### 2.1.3. Chronic Rosiglitazone Treatment Did Not Affect Acute Mortality during Ischemia/Reperfusion Injury

As a significant measure for hidden cardiotoxicity, we tested the mortality caused by I/R injury. The chi-square test did not show any significant differences between the four treatment groups illustrated in Figure 1. Figure 4 presents the mortality in percent. In the I/R vehicle group, two of the animals died during ischemia due to irreversible ventricular fibrillation. In the rosiglitazone-treated I/R group, two of the animals died. One died due to ventricular fibrillation in the ischemic period, and the other died due to bradycardia during reperfusion. Animals that died during the short recurrent I/R episodes of IPC were excluded from further evaluations and figures (IPC vehicle: six rats, IPC rosiglitazone: five rats). All animals survived in the IPC vehicle group. One rat died due to asystole during reperfusion in the IPC rosiglitazone group. These results show that rosiglitazone does not aggravate mortality in this model of I/R injury.

### 2.2. In Vitro Model of Simulated Ischemia/Reperfusion Injury

To uncover the hidden cardiotoxic properties of rosiglitazone in vitro, we performed I/R injury experiments in isolated adult rat cardiomyocytes (ARCMs), AC16, and differentiated AC16 cells (diffAC16s) (Figure 5).

#### 2.2.1. Rosiglitazone Treatment Increased Cell Survival of Adult Rat Cardiomyocytes with 0.3 μM in Simulated Ischemia/Reperfusion Injury

The in vitro assays are illustrated in Figure 5 and were designed to test the direct effect of rosiglitazone on cell survival under simulated ischemic conditions. With respect to the isolated primary cells shown in Figure 6a, the simulated ischemia/reperfusion (sI/R) with vehicle treatment caused significant cell death within three hours of simulated ischemia and two hours of reperfusion compared to the normoxic control. The cell viability was increased with 0.3 μM rosiglitazone in sI/R injury (Figure 6b), whereas none of the rosiglitazone concentrations affected the viability in a normoxic environment (Appendix A). These results indicate that rosiglitazone does not aggravate cell death in ARCMs within a clinically relevant dose range and that 0.3 μM rosiglitazone may promote cell survival.

#### 2.2.2. Rosiglitazone Treatment Increased the Viability of AC16 Cells with 0.1 and 0.3 μM Concentrations but Not of Differentiated AC16 Cells

Two human cell lines were exposed to 16 h of simulated ischemia and 2 h of reperfusion to detect hidden cardiotoxicity. As shown in Figure 7 and Figure 8, the viability of the AC16s and diffAC16s was reduced with sI/R and vehicle treatment compared to the normoxic control. However, the reduced cell survival with simulated ischemia was reversed with 0.1 µM and 0.3 µM rosiglitazone treatment in the AC16 cell line (Figure 7). Following the differentiation of the AC16 cells, no such significant increases (*p* < 0.05) in viability were observed (Figure 8). Appendix A shows a decrease in AC16 cell survival with 10 µM rosiglitazone treatment under normoxic conditions. The results obtained for the diffAC16 cell line under normoxic conditions show no change with rosiglitazone treatment compared to the vehicle group (Appendix A). Together, these results illustrate a difference between the two cell lines in terms of response to rosiglitazone and sI/R.

## 3. Discussion

In this article, we showed that rosiglitazone does not aggravate infarct size, arrhythmia score, or mortality during I/R injury in rats. We also demonstrated that rosiglitazone abolishes the antiarrhythmic effects of IPC during I/R injury, whereas its infarct-size-limiting effects are not compromised. Our in vitro data, wherein sI/R injury was not deteriorated by rosiglitazone, support the neutral results of our in vivo experiments. This is the first demonstration that chronic administration of rosiglitazone does not show major hidden cardiotoxic effects in myocardial I/R injury models.

In our present study, chronic rosiglitazone treatment did not influence acute I/R-induced infarct size or the infarct-size-limiting effect of IPC, which is seemingly in contrast to the current literature. Several studies have reported a reduced infarct size with 7 to 14 days of chronic rosiglitazone treatment of 3 to 5 mg/kg/day in rats [22,23,24,25,26]. However, in three of these studies, the last dose was administered one hour before the surgery [22,23,24], and the other two papers did not provide information on the timing of the last dose [25,26]. Therefore, due to their experimental setup, a direct cardioprotective effect could explain the reduction in infarct size, which is supported by the infarct-size-limiting effect of acute rosiglitazone treatment [24,27,28,29]. In addition, all experiments with chronic rosiglitazone treatment in rats used high doses, which resulted in peak plasma concentrations [30,31] exceeding those in humans [32,33,34,35,36,37,38,39,40,41]. The average peak plasma concentration of rosiglitazone resulting from single-dose administration of 1 mg/kg in rats is greater than the peak plasma concentration in humans after single-dose administration of 4 or 8 mg doses [42,43]. Our dose of 0.8 mg/kg/day more accurately resembles the actual human conditions than those used in other chronic rosiglitazone treatment studies. One study in rabbits that applied chronic rosiglitazone treatment at doses of 0.5 mg/kg/day (similar to our dose) reported a decreased infarct size [44]. However, this workgroup also showed an infarct in the sham-operated group similar in size to that of rosiglitazone-treated animals exposed to I/R. In conclusion, hidden cardiotoxicity of chronic rosiglitazone treatment does not manifest as an increase in infarct size; however, a direct effect of the drug in an acute setting cannot be excluded.

Our present results demonstrate that rosiglitazone did not increase the arrhythmia score through ventricular premature beats, ventricular tachycardias, or ventricular fibrillations during I/R; instead, it abolished the antiarrhythmic effect of IPC. A similar study suggested that the severity of ventricular fibrillations might have been worse with rosiglitazone treatment in animals protected by IPC. In this paper, the mortality of pigs that received IPC and were exposed to ischemia until ventricular fibrillation was increased by rosiglitazone [45]. A number of clinical studies have investigated whether antidiabetics abrogate cardioprotection by modulating the underlying intracellular signaling pathways [4]. Similar to our experiments with rosiglitazone, the antidiabetic repaglinide abolished IPC-induced protection in patients with coronary artery disease during exercise stress testing, which manifested as pronounced ischemic ECG signs, resulting in increased arrhythmia-susceptibility [46,47]. With respect to direct arrhythmia risk, several studies of acute rosiglitazone treatment revealed a decreased time interval from left anterior descending coronary artery occlusion to onset of ventricular fibrillation [27,45,48,49]. Additionally, Palee et al. described an increased arrhythmia score and incidence of spontaneous ventricular fibrillation after I/R in rats and pigs [27,48]; however, these findings can be attributed to the acute setting. In contrast, chronic rosiglitazone treatment significantly reduced arrhythmogenesis in rats during I/R [22], which also might be explained by the presence of active metabolites and possibly the considerably higher doses. Further results with a similar dose to ours (0.5 mg/kg/day) and chronic treatment in rabbits show that rosiglitazone reduces the incidence of I/R-induced arrhythmias [44]; however, this study failed to present the dataset underlining the statement. Lee et al. treated diabetic hypertensive rats with rosiglitazone for 14 days and found that rosiglitazone increases arrhythmogenic potential in isolated ventricular myocytes without I/R [50]. These findings further strengthen the arrhythmic risks of rosiglitazone treatment by revealing proarrhythmic properties in a diseased heart. The arrhythmogenic potential of rosiglitazone also might have been observed in clinical settings. A retrospective pharmacovigilance analysis reported that rosiglitazone is the second most commonly reported medication for ventricular fibrillation [51]; however, it might not manifest as an increased hazard of sudden cardiac death [52].

Here, we have demonstrated that chronic rosiglitazone treatment does not increase mortality during acute I/R injury in rats. As a surrogate marker, similar findings in pigs subjected to ischemia until ventricular fibrillation showed that acute rosiglitazone treatment did not influence the success rate of defibrillation [45]. However, most of the data are ambiguous with respect to mortality, e.g., acute rosiglitazone treatment increased the mortality rate in rats during I/R [27]. Mice treated with 20 mg/kg/day rosiglitazone for three days prior to their MI and then for the following eleven days had a decreased post-MI survival rate [53], whereas chronic eight-week treatment with 3 mg/kg/day rosiglitazone initiated during reperfusion aggravated post-MI mortality in rats [54]. Several clinical observations reported no increased risk for cardiovascular or all-cause death with chronic rosiglitazone treatment [16,18,20,21,55]. Florez et al. described a decrease in cardiovascular death in the VAD trial [17]. A study following a patient population with established ischemic cardiovascular disease compared pioglitazone and rosiglitazone, revealing an increased risk of MI but no adversely altered overall mortality [56]. In the Bypass Angioplasty Revascularization Investigation 2 Diabetes (BARI 2D) trial, patients with angiographically confirmed coronary heart disease were started on rosiglitazone or non-thiazolidinedione therapy regimens. During 4.5 years of follow-up, rosiglitazone treatment resulted in insignificantly changed rates of nonfatal MI among this high-risk patient population [55]. These clinical observations align with our experimental results, supporting the conclusion that chronic rosiglitazone treatment does not deteriorate the mortality rate of MI. In contrast, Nissen et al. found an increased risk of cardiovascular death in 2007 [19]. However, their updated meta-analysis in 2010 did not find significance [57]. In further reports comparing pioglitazone and chronic rosiglitazone treatment, an association with a significantly increased death rate upon rosiglitazone treatment was identified [58,59]. Interestingly, there was no increased risk for all-cause mortality in a more differentiated approach whereby pioglitazone and rosiglitazone were independently compared to metformin. However, when comparing the two thiazolidinediones, rosiglitazone aggravated the death rate [60]. In conclusion, our findings are consistent with the relevant clinical observations, as hidden cardiotoxicity of rosiglitazone does not manifest as increased mortality.

In our in vitro experiments, clinically relevant doses of rosiglitazone did not cause cell death in primary ARCMs in normoxia or sI/R. During sI/R, 0.3 μM rosiglitazone even increased cell survival. The applied dose range of 0.1 to 10 μM corresponds to the human peak plasma concentration range of 0.8 to 1.96 μM (averaged data from humans [32,33,34,35,36,37,38,39,40,41]) and our in vivo peak plasma concentration of approximately 8.2 μM [42,43]. Our results are consistent with data obtained from isolated mitochondria from ARCMs showing no change in reactive oxygen species or membrane potential due to 10 to 50 μM concentrations of rosiglitazone [48]. However, in neonatal rat and adult mouse cardiomyocytes, reduced apoptosis markers were associated with 10 μM rosiglitazone concentration [61,62]. Here, we showed the effect of rosiglitazone treatment on human cardiac cell lines exposed to sI/R for the first time. AC16 cells showed improved cell survival resulting from 0.1 and 0.3 μM rosiglitazone concentrations, whereas diffAC16s did not show improved viability at any concentration. Based on previous results of our workgroup [63], this may be attributed to metabolic changes upon differentiation, resulting in increased hypoxia sensitivity. Mersmann et al. described a concentration- and reperfusion-time-dependent increase in cell survival at a concentration range well over human peak plasma concentration (10–100 μM rosiglitazone) for the rat H9c2 cell line [64]. Because cells in all in vitro experiments were subjected to the direct action of rosiglitazone, the obtained results may not be directly comparable to the findings of our in vivo experiments with chronic rosiglitazone treatment. However, they support our results, showing no increase in infarct size due to rosiglitazone treatment.

In conclusion, this is the first demonstration that chronic administration of rosiglitazone does not show major hidden cardiotoxic effects in models of myocardial I/R injury. Because rosiglitazone inhibits the antiarrhythmic effects of IPC and is still available in the US, pharmacovigilance or clinical studies could reveal the clinical relevance of our findings in patients receiving cardioprotective interventions.

## 4. Materials and Methods

### 4.1. Ethical Considerations

This study was conducted based on the 3Rs rule, which incorporates replacement, reduction, and refinement. The experiments were conducted in compliance with the Guide for the Care and Use of Laboratory Animals published by the US National Institutes of Health (NIH publication No. 85–23, revised 1996) and to the EU Directive (2010/63/EU). This study was approved by the Animal Ethics Committee of Semmelweis University (Budapest, Hungary) and the National Scientific Ethical Committee on Animal Experimentation (Budapest, Hungary) and was authorized by the Food Chain Safety and Animal Health Directorate of the Government Office for Pest County, Hungary (PE/EA/1784-7/2017).

### 4.2. Sources of Chemicals

The majority of the chemicals were purchased from Sigma (St. Louis, MO, USA), including hydroxyethylcellulose (#09368), Evans blue dye (#E2129), triphenyltetrazolium chloride (#T8877), HEPES buffer (#H3375), dimethyl-sulfoxide (DMSO #D5879 or #D2650), 2-Deoxy-d-glucose (#D8375), and laminin (#L2020). Additional chemicals used in experiments were MgSO_4_ (Reanal, Budapest, Hungary, #20341), collagenase II (Biochrom GmbH, Berlin, Germany, #c2-22), fetal bovine serum (FBS, EuroClone, Pero MI, Italy, #ECS0180L), M199 (Lonza, Verviers, Belgium, #BE12-117F), bovine serum albumin (BSA, Santa Cruz Biotechnology, Santa Cruz CA, USA, #sc-2323), calcein AM (PromoCell GmbH, Heidelberg, Germany, #PK-CA707-80011-3), Dulbecco’s phosphate-buffered saline (DPBS, Gibco, Grand Island New York, NY, USA, #14080-055), heparin (Merck, Darmstadt, Germany, #375095), rosiglitazone (MedChemExpress, HY-14600), and pentobarbital (Produlab Pharma, Raamsdonksweer, The Netherlands, #17F015).

### 4.3. In Vivo Ischemia/Reperfusion Injury Study

#### 4.3.1. Animal Handling and Surgery Protocol

For the in vivo experiments, one-month-old male Wistar rats weighing 72–124 g were treated with 0.8 mg kg^−1^ day^−1^ rosiglitazone or vehicle (1% hydroxyethylcellulose) by oral gavage for 28 ± 1 days. The body-weight-development during the 28 ± 1 days is depicted in Appendix A. The animals arrived at the animal facility one week before the treatment started to allow them to get acclimatized with their surroundings. Cage-side visual observation for potential discomfort of the animals was routinely performed by trained animal keepers. The last dose was administered the day before the start of the surgery protocol. Thus, we focused on the chronically changed phenotype upon repeated treatment with a PPARγ transcription factor agonist because rosiglitazone has a half-life of 3 to 4 h [65]. According to the formula described by Reagan-Shaw et al. [66], the maximum human dose (8 mg daily) showing cardiovascular side effects in clinical studies [19] was converted to the applied rosiglitazone dose:HED (mg × kg^−1^) = Animal dose (mg × kg^−1^) × (rat Km)/(human Km)

To calculate the animal dose, the HED (human equivalent dose, 8 mg 60 kg^−1^ for an average 60 kg adult) was divided by the ratio of the average rat correction factor (rat Km = 6) and the average human correction factor (human Km = 37). Potential loss due to pharmacokinetics is negligible, as rosiglitazone has a bioavailability of 99% [65].

During the chronic treatment period, the animals were housed in a humidity-controlled room under a 12 h light/dark cycle with temperatures of 22 ± 2 °C. Laboratory chow and drinking water were available to the animals ad libitum. After 28 ± 1 days, the groups of animals treated with rosiglitazone or vehicle weighing 235–372 g received an intraperitoneal injection with 60 mg kg^−1^ pentobarbital as induction of anesthesia. Half-dose pentobarbital injections were repeated if the pedal reflex reoccurred to ensure deep surgical anesthesia. The body surface electrocardiogram (ECG), the heart rate (Appendix A) and the mean arterial blood pressure (MAP; Appendix A) were monitored throughout the experiments using standard limb leads and the cannulated right carotid artery (AD Instruments, Bella Vista, Australia), respectively. Arterial access was also used for fluid supplementation with saline containing 10 IU kg^−1^ heparin. To maintain the core body temperature at a physiological level, the animals were placed on a heating pad (Harvard Apparatus, Holliston, MA, USA). Rats were ventilated with 6.2 mL kg^−1^ room air at 69 ± 3 breath min^−1^ using a rodent ventilator (Ugo-Basile, Gemonio, Italy). The 0 min of the experiment was appointed after successfully placing a 5-0 prolene suture (Ethicon, Johnson & Johnson, Budapest, Hungary) around the left anterior descending (LAD) coronary artery. Intraperitoneal heparin injections of 100 IU kg^−1^ were administered at 35, 65, and 185 min of the experiments.

The experimental design and surgery protocol are depicted in Figure 1. The rosiglitazone-treated and the vehicle-treated group each comprised 34 animals. On the day of surgery, both treatment groups were divided into two surgery protocols comprising I/R with or without IPC. To assign more animals to the higher-mortality groups, directed randomization was used: I/R + vehicle group (*n* = 14), I/R + rosiglitazone group (*n* = 15), IPC + vehicle group (*n* = 20), IPC + rosiglitazone group (*n* = 19). For the in vivo experiments, the total number of animals was 68. IPC was elicited by three cycles of brief 5 min LAD occlusion and 5 min reperfusion before I/R, whereas the I/R group was subjected only to 30 min of LAD occlusion. All animals were exposed to 120 min of reperfusion following the ischemic period [67,68]. Successful LAD occlusion was confirmed by ST-segment elevation or depression, the appearance of arrhythmias, and pallor of myocardial regions distal to the occlusion site.

#### 4.3.2. Infarct Size Measurement

Animal hearts were excised after 120 min of reperfusion and subsequently perfused in Langendorff mode for 2 min with oxygenated Krebs–Henseleit solution (in mM: NaCl 118, KCl 4.7, MgSO_4_ 1.2, CaCl_2_ 1.25, KH_2_PO_4_ 1.2, NaHCO_3_ 25, and glucose 11) at 37 °C to remove blood from the tissue. After reocclusion of the LAD, the total area exposed to ischemia was negatively stained with Evans blue dye through the ascending aorta. To distinguish the AAR from the infarcted tissue, 2 mm thick slices were cut and incubated in 1 % triphenyltetrazolium chloride for 14 min at 37 °C. The slices were weighed, scanned, and individually evaluated by two blinded investigators. Planimetric analyses were performed with InfarctSize 2.4 b software (Pharmahungary Group, Budapest, Hungary). The infarct size is expressed as a percentage of the AAR, a ratio of the total left ventricular area (100%). The area ratios were then normalized to the weight of each slice. Finally, the sum of the weight-adjusted areas was divided by the weight of the whole heart and multiplied by 100.

#### 4.3.3. Arrhythmia Analysis

For the arrhythmia analysis, blinded investigators assigned a score to the arrhythmias occurring during the 30 min of ischemia and the first 15 min of reperfusion. This evaluation was performed according to the Lambeth conventions and quantified as previously described by Curtis and Walker using score A [69,70].

#### 4.3.4. Mortality Analysis

Animals included in the mortality analysis died either due to irreversible VF, pulseless electrical activity, or bradycardia (<150 BPM), accompanied by hypotension (MAP < 15 mmHg). Animals that died as a result of IPC-induced arrhythmias (11 animals) were excluded, as well as all other iatrogenic causes of death (6 animals).

### 4.4. In Vitro Simulated Ischemia/Reperfusion Injury Study

#### 4.4.1. ARCM Isolation Protocol

Male Wistar rats weighing 170–200 g were anesthetized by intraperitoneal pentobarbital injection (60 mg kg^−1^). Each animal was subjected to heparinization (500 IU kg^−1^) via the femoral vein. The hearts were excised and placed into 0 °C Krebs–Henseleit solution. Subsequently, the aorta was cannulated in the Langendorff system, and the heart was perfused retrogradely for 2–4 min with oxygenated Krebs–Henseleit solution to wash out the blood. The perfusate was then switched to a digestive Krebs solution containing collagenase II (8000 U/mL) for 30 min. Following this perfusion, the ventricles were excised from the heart, chopped into small pieces, and placed into a Falcon tube containing the previously mentioned digestive solution. Prior to filtrating the cell suspension, the digestion continued for 10 min. When the cells in the suspension had successfully pelleted, the pellet was washed 2–3 times with Krebs–Henseleit solution while gradually titrating up the CaCl_2_ concentration to 1 mM [71]. The isolated cells were plated in laminin-coated 24-well plates (Thermo Fisher Scientific, Waltham, MA, USA) with a seeding density of 7500 cells/well and a proliferative medium (M199, 1 mL/well) containing 5% fetal bovine serum (FBS). The plates were incubated for three hours, and the medium was replaced by growth medium without serum. One animal heart was used for *n* = 1. The total number of animals used for ARCM isolations was 143. Exclusion criteria were (I) low viability (<50%) on the day of isolation and (II) low viability on day one of culturing (<50%) in accordance with our previous studies [7,72]. The number of excluded isolations due to criterion (I) was 66 and 18 due to criterion (II). Thus, we used 47 isolations for the experiments.

#### 4.4.2. Human Cardiac Cell Line AC16 Maintenance and Differentiation

The human cardiac myocyte AC16 cell line was obtained from Merck (Master cell bank passage 4, Lot: RD1606008; SCC109). All experiments were performed within 10 passages of the working cell bank. For cultivation, the cells were kept in growth medium (DMEM/Nutrient Mixture F-12 (04-687F/U1, Lonza), 12.5 V/V% FBS (ECS0180L, EuroClone or 35-079-CV, Corning, Corning, NY, USA), 100 uU/mL penicillin, 100 ug/mL streptomycin, and 25 ng/mL amphotericin B (30-004-CI, Corning)) at 37 °C in a humidified atmosphere of 5% CO_2_. Prior to dividing the subcultures, cells were maintained until 70% confluence. For differentiation of AC16 cells, growth medium was changed to differentiation medium with 2% FBS supplemented with 10 nM ATRA and 1× insulin-transferrin-selenite supplement (DMEM/Nutrient Mixture F-12 (04-687F/U1, Lonza), as well as 2 V/V% FBS (ECS0180L, EuroClone or 35-079-CV, Corning), 100 uU/mL penicillin, 100 ug/mL streptomycin and 25 ng/mL amphotericin B (30-004-CI, Corning), 1× ITS (I3146, Sigma), and 10 nM ATRA (R2625, Sigma)) [63]. AC16 and diffAC16s were seeded on 96-well plates with 2 × 104 and 1 × 104 seeding density, respectively.

#### 4.4.3. Simulated Ischemia/Reperfusion and Normoxia Conditions

The study protocols for the in vitro cell culture experiments are illustrated in Figure 5. The cells were plated for 24 h prior to the exchange of growth media with growth media containing vehicle (1 μL DMSO/ 1 mL medium or normoxic or hypoxic solution) or rosiglitazone in increasing concentrations (0.1 μM, 0.3 μM, 1 μM, 3 μM, and 10 μM). This dose range was extrapolated from a simple conversion of the applied in vivo dose:0.8 mg/kg = 0.8 mg/L = 0.8 μg/mL

The 0.8 μg/mL concentration can be translated to 2.2 mM, which was used as a point of reference for the rosiglitazone dose range in the in vitro study protocol. Using a dose range in cell culture experiments is a common practice to cushion interspecies differences.

Following incubation for one hour in a CO_2_ incubator (Scancell—Labogene, Lynge, Denmark), the growth medium was replaced for three hours (ARCMs) or 16 h (AC16s, diffAC16s) with either a normoxic solution (in mM: NaCl 125, KCl 5.4, NaH_2_PO_4_ 1.2, MgSO_4_ 1.3, CaCl_2_ 1, glucose 15, taurine 5, creatine-monohydrate 2.5, and BSA 0.1%, pH7.4) in a CO_2_ incubator (normoxia groups) or with hypoxic solution (in mM: NaCl 119, KCl 5.4, NaH_2_PO_4_ 1.2, MgSO_4_ 1.3, HEPES 5, MgCl 0.5, CaCl_2_ 0.9, Na-lactate 20, and BSA 0.1%, pH6.4) in a three-gas (95% N_2_ and 5% CO_2_) incubator (sI/R groups, Panasonic Heathcare Co., Ltd., Gunma, Japan). During this period of normoxia or simulated ischemia, the cells were exposed to the previously mentioned concentrations of rosiglitazone or vehicle. Finally, both normoxic and sI/R-conditioned cells were replenished in normoxic growth medium in the CO_2_ incubator for 2 h with the corresponding rosiglitazone or vehicle treatment [71,73,74]. Eight ARCM isolations were used for eight successful normoxia plates, and 39 isolations were used for eight successful sI/R plates. One isolation was used for a normoxia and a sI/R plate. For the AC16 cells, 7 of 14 plates were successful for the normoxic group, and 4 of 21 plates were successful for the sI/R plates. The diffAC16s had five successful normoxia plates out of six plates and four successful sI/R plates out of eight. SI/R plates with no significant difference (*p* > 0.1 for ARCM and *p* > 0.05 for AC16s/diffAC16s) between the normoxia and the sI/R group were excluded. Furthermore, plates with low fluorescence intensity (RFU < 0.1 for ARCM and RFU < 0.3 for AC16), technical failures, or a coefficient of variation higher than 100 and lower than 0 were excluded.

#### 4.4.4. Viability Assay

After completing the study protocol shown in Figure 3, calcein staining was applied to assess the cell viability of the previously introduced cell cultures. The cells were washed with preheated 1% DPBS, and subsequentially, calcein dye (1 μM) was applied at room temperature in a dark chamber for 30 min. Finally, the calcein solution was washed with DPBS, and an unbiased evaluation was performed. To detect the fluorescence intensity of each well automatically, a Varioskan Lux multimode microplate reader (Thermo Fisher Scientific, Waltham, MA, USA) was used at 37 °C with an excitation wavelength of 490 nm and an emission wavelength of 520 nm. Relative fluorescence units % (RFU%) express the cell survival as an arbitrary unit. To assess the autofluorescence of rosiglitazone, all applied drug concentrations were measured in DPBS, and no signal was detected. Therefore, the viability assay results were not influenced by the treatment with rosiglitazone [71,72,73]. For the ARCMs shown in Figure 6a, the normoxia (N) + vehicle group was set to 100% relative fluorescent units (RFU), and all of the data were normalized to the averaged simulated ischemia (SI)+ vehicle group. As shown in Figure 6b, the SI + vehicle group was set to 100% RFU, and all data were normalized to the averaged SI + vehicle group. The AC16 cell line is shown in Figure 7, with the N + vehicle group set to 100% RFU and all data were normalized to the averaged N + vehicle group. For the diffAC16 cell line shown in Figure 8, the N + vehicle group was set to 100% RFU, and all data were normalized to the averaged medium-treated group.

### 4.5. Literature Search Methodology

We performed a systematic literature search to identify studies showing cardiotoxic or cardioprotective properties of rosiglitazone under ischemic conditions in humans in vivo and in cells. On 20 February 2022, our search string “((Rosiglitazone OR Avandia) AND (cardi* OR myocardi* OR heart OR cardiac OR cardio OR cardiology) AND ((Ischemia) OR (Reperfusion) OR (heart attack) OR (myocardial infarction) OR (Infarct))) NOT (review [Publication Type])” resulted in 289 hits. During further analysis, articles related to pioglitazone, non-related articles, reviews, comments, and articles with no abstract were excluded. Based on these criteria, 184 papers were excluded. Clinical studies were investigated to identify observations opposing or confirming whether rosiglitazone has cardiotoxic effects in humans. In vitro or in vivo experiments with endpoints different from ours were analyzed to identify relationships to our results. In studies with a similar setup, we identified differences and coherences to compare the results reported in the current literature to our results.

### 4.6. Statistical Analysis

The chi-Square test was applied to identify differences in the mortality percentages (Figure 4). Data presented in Figure 3 are expressed as the median with individual data points. Data presented in Figure 2, Figure 6a,b, Figure 7 and Figure 8 are expressed as the mean with standard error. The significance level was set to 5 % using the Kruskal–Wallis test with Dunn’s post hoc tests for multiple comparisons (Figure 2, Figure 3, Figure 6b, Figure 7 and Figure 8) or Mann–Whitney test (Figure 6a) to match the need of non-parametric datasets. The absence of normal distribution was confirmed by Shapiro–Wilk and Kolmogorov–Smirnov tests. Statistical analysis was performed using GraphPad Prism (version 6.0, GraphPad Software, San Diego, CA, USA).

## 5. Conclusions

Our experiments demonstrate that chronic administration of rosiglitazone does not show major hidden cardiotoxic effects in cellular and animal models of myocardial I/R injury. Because rosiglitazone inhibits the antiarrhythmic effects of IPC and is still available in the US, pharmacovigilance or clinical studies could reveal the clinical relevance of our findings in patients receiving cardioprotective interventions.

## 6. Limitations

Although we did not measure cardiac function or perform other histopathological or immunohistochemical analyses to further prove cardiotoxicity, this model is suitable for such measurements.

## Figures and Tables

**Figure 1 pharmaceuticals-15-01055-f001:**
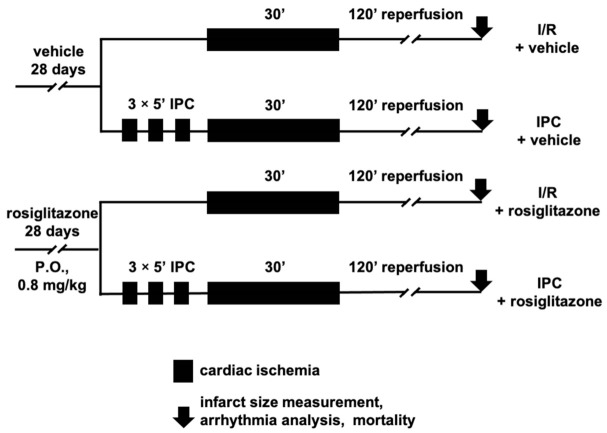
**In vivo experimental protocol of myocardial ischemia/reperfusion injury and ischemic preconditioning in rats.** Ischemia was implemented by occlusion of the left anterior descending coronary artery, and an electrocardiogram, body temperature, heart rate (Appendix A), and blood pressure (Appendix A) were recorded during the protocol. **IPC**: ischemic preconditioning; **I/R**: ischemia/reperfusion; **P.O.**: per os.

**Figure 2 pharmaceuticals-15-01055-f002:**
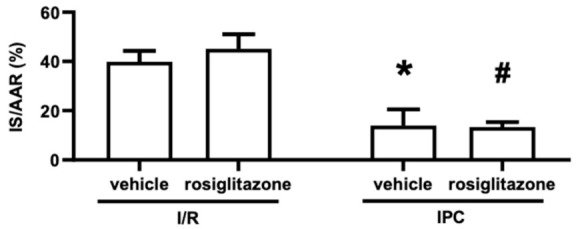
**Myocardial infarct sizes as % of the area at risk.** Results are presented as mean with standard error. Chronic rosiglitazone treatment does not aggravate infarct size of ischemia/reperfusion injury and does not interfere with cardioprotection by ischemic preconditioning. (Kruskal–Wallis, Dunn’s post hoc, * *p* < 0.05 vs; I/R + vehicle, # *p* < 0.05 vs. I/R + rosiglitazone, *n* = 9–10) **IPC**: ischemic preconditioning; **I/R**: ischemia/reperfusion; **IS/AAR**: infarct size/area at risk.

**Figure 3 pharmaceuticals-15-01055-f003:**
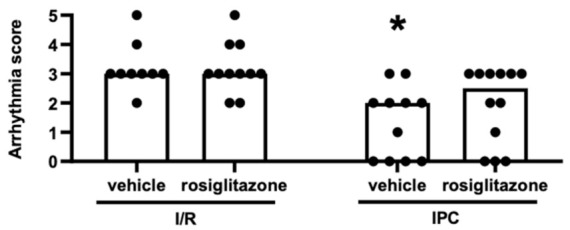
**Arrhythmia scores over the entire duration of the myocardial ischemic period and the first 15 min of reperfusion**. Results are presented as median with individual data points. Chronic rosiglitazone treatment does not increase the arrhythmia score during ischemia/reperfusion but abolishes the antiarrhythmic effect of ischemic preconditioning. (Kruskal–Wallis, Dunn’s post hoc, * *p* < 0.05 vs. I/R + vehicle, *n* = 11–12) **IPC**: ischemic preconditioning; **I/R**: ischemia/reperfusion.

**Figure 4 pharmaceuticals-15-01055-f004:**
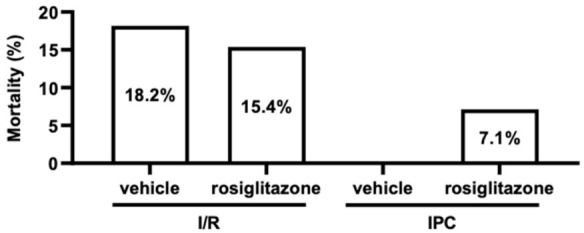
**Mortality.** Chronic rosiglitazone treatment does not influence mortality during ischemia/reperfusion injury, nor does it affect mortality in connection with ischemic preconditioning. (Percentage; chi-square test, *n* = 11–14). **IPC**: ischemic preconditioning; **I/R**: ischemia/reperfusion.

**Figure 5 pharmaceuticals-15-01055-f005:**
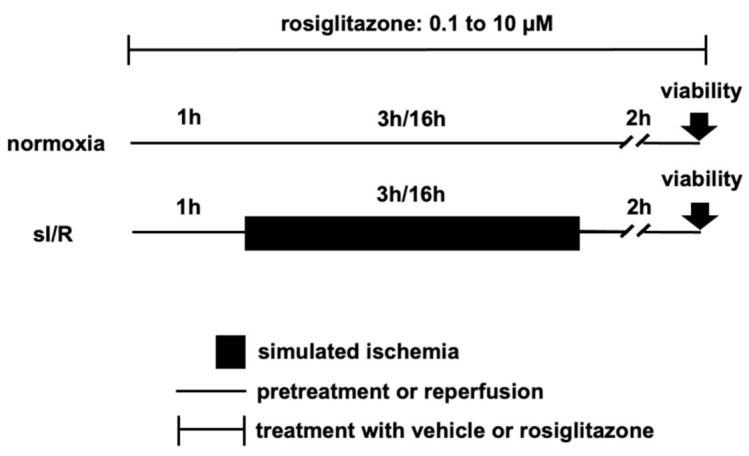
**Experimental protocols of simulated ischemia/reperfusion injury with different rosiglitazone concentrations in vitro.** ARCMs were exposed to three hours of simulated ischemia followed by 2 h of simulated reperfusion, whereas the immortal AC16 and differentiated AC16 cells were subjected to 16 h of simulated ischemia followed by 2 h of simulated reperfusion. **sI/R**: simulated ischemia/reperfusion.

**Figure 6 pharmaceuticals-15-01055-f006:**
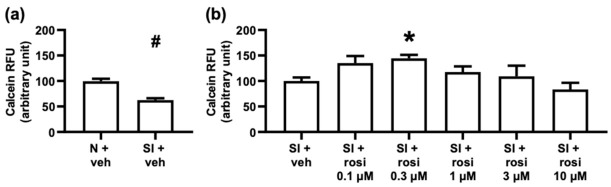
Cell viability under normoxic and simulated ischemic conditions with or without varying concentrations of rosiglitazone. Results are presented as mean with standard error. (**a**) Adult rat cardiomyocytes. The normoxia (N) + vehicle group was set to 100% relative fluorescent units (RFU), and all data were normalized to the averaged simulated ischemia (SI) + vehicle group. (Mann–Whitney test, # *p* < 0.05 vs. N + vehicle, *n* = 9, 38 technical replicates) (**b**) Adult rat cardiomyocytes. The SI + vehicle group was set to 100% RFU, and all of data were normalized to the averaged SI + vehicle group. (Kruskal–Wallis, Dunn’s post hoc, * *p* < 0.05 vs. SI + vehicle, *n* = 4–9, 12–40 technical replicates) (**a**,**b**) Adult rat cardiomyocytes. Rosiglitazone treatment increased the viability of adult rat cardiomyocytes at 0.3 µM during simulated ischemia. SI: simulated ischemia; N: normoxia; veh: vehicle.; rosi: rosiglitazone; RFU: relative fluorescence units.

**Figure 7 pharmaceuticals-15-01055-f007:**
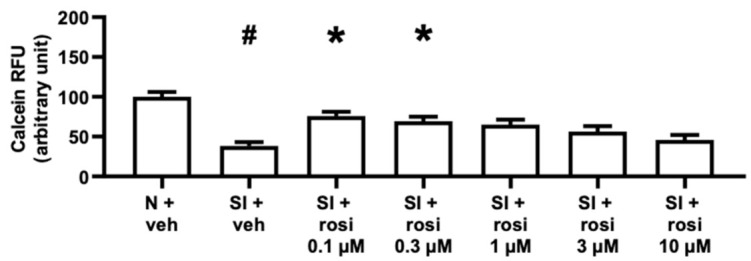
AC16 cell line viability under normoxic and simulated ischemic conditions with or without varying concentrations of rosiglitazone. Results are presented as mean with standard error. The N + vehicle group was set to 100% RFU, and all data were normalized to the averaged N + vehicle group. Rosiglitazone treatment increased the viability of AC16 cells at 0.1 µM and 0.3 µM during simulated ischemia. (Kruskal–Wallis, Dunn’s post hoc, * *p* < 0.05 vs. SI + vehicle, # *p* < 0.05 vs. N + vehicle, *n* = 4, 32–39 technical replicates). SI: simulated ischemia; N: normoxia; veh: vehicle; rosi: rosiglitazone; RFU: relative fluorescence units.

**Figure 8 pharmaceuticals-15-01055-f008:**
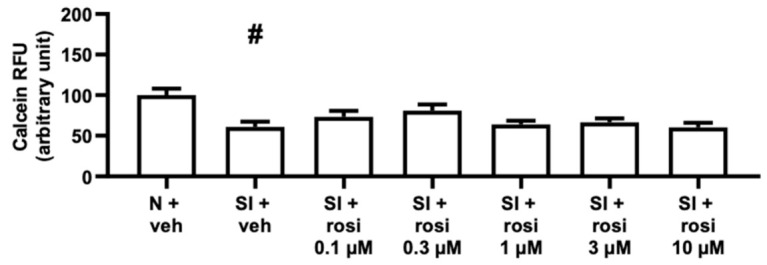
Differentiated AC16 cell line viability under normoxic and simulated ischemic conditions with or without varying concentrations of rosiglitazone. Results are presented as mean with standard error. The N + vehicle group was set to 100% RFU, and all data were normalized to the averaged medium-treated group. Rosiglitazone treatment did not aggravate cell death due to simulated ischemia in differentiated AC16 cells. (Kruskal–Wallis, Dunn’s post hoc, # *p* < 0.05 vs. N + vehicle. *n* = 4, 21–22 technical replicates). SI: simulated ischemia; N: normoxia; veh: vehicle; rosi: rosiglitazone; RFU: relative fluorescence units.

## Data Availability

Data is contained within the article and Appendix A.

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
