# Peer review of "Rosiglitazone Does Not Show Major Hidden Cardiotoxicity in Models of Ischemia/Reperfusion but Abolishes Ischemic Preconditioning-Induced Antiarrhythmic Effects in Rats In Vivo"

_pharmaceuticals, 2022, doi:10.3390/ph15091055_

Round 1

Reviewer 1 Report

This work indicated that rosiglitazone doesn’t show hidden cardiotoxicity in preclinical models of ischemia/reperfusion, but abolishes ischemic preconditioning-induced antiarrhythmic effects in rats in vivo. There are some issues in this manuscript that should be addressed as follows:

  • The approval codes of the animal ethics committee of the Semmelweis University, Budapest, Hungary, and the National Scientific Ethical Committee on Animal Experimentation should be mentioned.
  • How did you know that the animals were acclimatized?
  • The exact number of animals used in this study should be mentioned.
  • A reference for the method of induction of ischemia/reperfusion injury should be provided.
  • A reference for the ARCM isolation protocol should be provided.
  • A reference for the simulated ischemia/reperfusion and normoxia conditions should be provided.
  • A reference for viability assay should be provided.
  • A major defect in this study is lack of histopathological and immunohistochemical examination of the cardiac tissues to confirm the presence or absence of cardiotoxicity. Also, the cardiac function tests should be measured.
  • Page 13 Line 510: The sentence “figure 3. Is” should be replaced with “figure 3. was”.
  • A collective diagram summarizing the main findings of this study is recommended.
  • The conclusion is not sufficient. The possible clinical significance of the results of the present study should be mentioned.
  •  The manuscript should be revised by English-naïve speaker to improve the quality of the language.
  • The manuscript should be checked regarding grammatical errors and plagiarism.

Reviewer 2 Report

Manuscript entitled “Rosiglitazone does not show hidden cardiotoxicity in preclinical models of ischemia/reperfusion, but abolishes ischemic preconditioning-induced antiarrhythmic effects in rats in vivo” investigates an interesting scientific aspect of the hidden cardiotoxicity of rosiglitazone in the presence of both myocardial ischemia/reperfusion and cardioprotective ischemic preconditioning. However, there are some concerns that need to be addressed by authors as well as corrections before the publication.

General remarks

Term preclinical (I/R model) frequently used in the whole manuscript is unnecessary and should be avoided in general. It is well-know that every (animal) model is preclinical. The word model itself emphasizes the preclinical character therefore the term preclinical is redundant and should be avoided (state only as I/R model or animal model).

Please do not use the full stop (period) following the min as in Line 384: Animal hearts were excised after 120 min. of reperfusion or Line 390: chloride for 14 min. at 37 °C. The slices were weighed… this is frequently repeated through the whole text, please correct.

Title

Line 2-4: The title is written with the period at the end. Please correct this grammatical error.

Introduction

Introduction is well-written from the scientific point of view. However, it is a bit tiresome to read because of long complex sentences which tend to loss their point in the end. I strongly suggest rephrasing the Introduction in order to become much more acceptable for the potential readers. Finally, the point of each Introduction is to keep reader’s focus and not to be too difficult to understand.

Grammatical/melodic errors

Lines 40-41: Sentence Hidden cardiotoxicity of a drug may further deteriorate injury cell signaling activated by I/R injury or cardiovascular comorbidities such as hyperlipidemia or diabetes. Please rephrase this sentence, you have the word injury doubled-written.

Line 49: Preclinical drug safety testing focus on the direct toxicity and adverse drug effects in…please correct to focuses

Materials and methods

Line 342-343: Sentence For the in vivo experiments, male Wistar rats of 72-124 g were treated with 0.8 mg kg- 1 day-1 rosiglitazone or vehicle (1% hydroxyethylcellulose) by oral gavage for 28 ± 1 days

·        question 1: How old are the rats at the begging of the experiment? According to their weights they are approximately 1 month old?

·        question 2: I have a concern regarding the oral gavage lasting for 28 days. How did you perform every-day gavage for such a long period of time? It is well-known that animals hardly endure a seven week every-day gavage….so 28 days seems like a rather long period.

Line 509: The Chi-Square test was applied to identify differences in the mortality percentages (figure 4.). Data presented in figure 3. Is shown as median with individual data points. Data presented in figures 2., 6. (a), 6. (b), 7., and 8. Are shown as mean with standard error. The significance level was set at 5 % using the Kruskal-Wallis test with Dunn´s post hoc. Pleases correct numerous grammatical errors in this paragraph.

Please specify what is the meaning of AAR..for example: The myocardial area at risk (AAR) is defined by the ischemic proportion of the myocardium after coronary occlusion and reflects the potential size of the myocardial infarction.

Results

The following lines i.e. Figures should be included in the materials and methods section, not in Results:

·        Line 84-89: Figure 1. Experimental protocol of myocardial ischemia/reperfusion injury and ischemic preconditioning in rats in vivo.

·        Line 145-150: Figure 5. Experimental protocols of simulated ischemia/reperfusion injury with different rosiglitazone concentrations 147 in vitro.

Line 106: Figure 2. Myocardial infarct sizes as % of the area at risk (AAR). The used AAR abbreviation cannot be introduced in the Figure caption. Please insert the AAR abbreviation into the 4.3.2. Infarct Size Measurement section or at the first place in the text were it is mentioned.

5. Conclusion

Line 518: Sentence This is the first demonstration that chronic administration of rosiglitazone does not 518 show hidden cardiotoxic effects in preclinical models of myocardial I/R injury, however, it interferes with the antiarrhythmic effects of IPC that may have clinical relevance in 520 terms of its hidden cardiotoxicity. This sentence is copy-pasted in the various parts of the manuscript. Please rephrase and not repeat this state at the same manner all over again.
